# Identification of Food Spoilage Fungi Using MALDI-TOF MS: Spectral Database Development and Application to Species Complex

**DOI:** 10.3390/jof10070456

**Published:** 2024-06-28

**Authors:** Nolwenn Rolland, Victoria Girard, Valérie Monnin, Sandrine Arend, Guillaume Perrin, Damien Ballan, Rachel Beau, Valérie Collin, Maëlle D’Arbaumont, Amélie Weill, Franck Deniel, Sylvie Tréguer, Audrey Pawtowski, Jean-Luc Jany, Jérôme Mounier

**Affiliations:** 1bioMérieux, R&D Microbiologie, Route de Port Michaud, F-38390 La Balme les Grottes, France; nolwenn.rolland@univ-brest.fr (N.R.); victoria.girard@biomerieux.com (V.G.); valerie.monnin@biomerieux.com (V.M.); sandrine.arend@biomerieux.com (S.A.); guillaume.perrin@biomerieux.com (G.P.); rachel.beau@biomerieux.com (R.B.); valerie.collin@biomerieux.com (V.C.); maelle.darbaumont@biomerieux.com (M.D.); 2Univ Brest, INRAE, Laboratoire Universitaire de Biodiversité et Écologie Microbienne, F-29280 Plouzané, France; damien.ballan@univ-brest.fr (D.B.); amelie.weill@univ-brest.fr (A.W.); franck.deniel@univ-brest.fr (F.D.); sylvie.treguer@univ-brest.fr (S.T.); audrey.pawtowski@univ-brest.fr (A.P.); jean-luc.jany@univ-brest.fr (J.-L.J.); 3Univ Brest, UBO Culture Collection, F-29280 Plouzané, France

**Keywords:** filamentous fungi, yeasts, cryptic species, *Aspergillus*, *Fusarium*, *Mucor*, *Colletotrichum*

## Abstract

Fungi, including filamentous fungi and yeasts, are major contributors to global food losses and waste due to their ability to colonize a very large diversity of food raw materials and processed foods throughout the food chain. In addition, numerous fungal species are mycotoxin producers and can also be responsible for opportunistic infections. In recent years, MALDI-TOF MS has emerged as a valuable, rapid and reliable asset for fungal identification in order to ensure food safety and quality. In this context, this study aimed at expanding the VITEK^®^ MS database with food-relevant fungal species and evaluate its performance, with a specific emphasis on species differentiation within species complexes. To this end, a total of 380 yeast and mold strains belonging to 51 genera and 133 species were added into the spectral database including species from five species complexes corresponding to *Colletotrichum acutatum*, *Colletotrichum gloeosporioides*, *Fusarium dimerum*, *Mucor circinelloides* complexes and *Aspergillus* series *nigri.* Database performances were evaluated by cross-validation and external validation using 78 fungal isolates with 96.55% and 90.48% correct identification, respectively. This study also showed the capacity of MALDI-TOF MS to differentiate closely related species within species complexes and further demonstrated the potential of this technique for the routine identification of fungi in an industrial context.

## 1. Introduction

The fungal kingdom is estimated to encompass between 2.2 to 3.8 million species, making it one of the widest groups on Earth [1]. This large group includes diverse eukaryotic microorganisms such as yeasts and filamentous fungi [2]. Those can have either positive or negative impacts on human activities. Indeed, fungi can produce a wide range of pharmaceutical products, enzymes and organic acids [2]. They are also major actors in food and beverage industries due to their ability to modify and improve the organoleptic and nutritional properties of food products from animal and plant origins as well as their ability to increase food shelf-life through fermentation [3,4,5]. They are involved, for instance, in the manufacturing process of soy sauce, miso, tempeh, mold-ripened cheeses, fermented sausages, bread, kombucha, beer, wine and various spirits.

Conversely, due to their ability to colonize a very large diversity of food raw materials and processed foods along the food chain, fungi are also major contributors to global food losses and waste which represent ~1.3 billion tons each year [6]. As an example, Davies et al. (2021) estimated that fungi were involved in up to 20% of global crop yield losses with at least 125 million tons of the five most cultivated crops lost each year because of fungal growth [6]. Moreover, Pitt and Hocking [5] estimated that fungal spoilage was responsible for 5–10% of food losses and waste. It is also worth mentioning that fungal spoilage leads to substantial financial losses [7,8] and the waste of natural resources (land, water and greenhouse gas emission) and contributes to food insecurity worldwide. 

*Aspergillus*, *Penicillium* and *Fusarium* are the main genera involved in food spoilage. Several species of these genera are mycotoxin producers which represent a major hazard for human and animal health [9,10]. Indeed, of the more than 300 mycotoxins that have been identified so far, 6 of them, namely aflatoxins, fumonisins, ochratoxins, patulin, trichothecenes and zearalenone, are regularly found in food, leading to unpredictable and ongoing food safety problems at a global scale [11]. Furthermore, some species within the genera *Mucor*, *Aspergillus* and *Penicillium*, among others, are also responsible for opportunistic infections in immunocompromised patients [12,13]. In this context, the rapid and accurate identification of fungi at the species level is crucial to ensure food safety and quality [14,15,16].

Traditionally, fungal identification was performed using phenotypic approaches, i.e., morphological and biochemical characteristics [17,18]. However, those approaches are tedious, time-consuming, may be prone to misidentification and require high expertise [7,8]. Over the last two decades, DNA barcoding, which relies on the sequencing of one or more standardized short DNA regions, has drastically modified the ability to identify fungal species [19]. The internal transcribed spacer (ITS) region of the nuclear ribosomal DNA (rDNA) is, for instance, a highly polymorphic region that is considered the universal barcode marker for fungi [20]. Hence, DNA barcoding is nowadays considered as the gold standard because of its reliability and accuracy but remains at the same time an expensive and tedious method requiring special skills and knowledge to be applicable in routine examination in an industrial setting [19,21].

In recent years, matrix-assisted laser desorption ionization–time of flight mass spectrometry (MALDI-TOF MS) has emerged as a valuable, rapid, cost-effective and reliable asset for microorganism identification [22,23]. This technique was initially applied to bacterial and yeast species before expanding in recent years to filamentous fungus identification [21,24,25,26]. MALDI-TOF MS is now commonly used for routine microbial identification in clinical and industrial microbiology laboratories. It relies on the rapid and precise analysis of biomolecules such as proteins, peptides, nucleic acids and lipids yielding a specific spectral signature which can then be identified by comparison to a reference spectral library [7,25,27]. Nevertheless, the lack of available spectra in commercialized databases, particularly for fungi, is still a concern [18,21,28]. Spectral database implementation for food-relevant species is therefore necessary to keep this technique up to date in the face of current challenges including the addition of novel target species or fungal taxonomy updates and changes. Concerning the latter aspect, MALDI-TOF MS has been successfully applied to discriminate species within species complexes [26,29,30]. A species complex is defined as a cluster of closely related species [31] which may include cryptic species [21]. Species within a species complex are difficult to distinguish using traditional phenotypic methods and may require the analysis of several specific genes [32]. Despite their close phylogenetic relatedness, these species may exhibit significant differences in their physiology, metabolic or ecological traits [33] and therefore could have a positive or negative impact on human activities as mentioned above. As an example, Quéro et al. [26] were able to correctly identify species of the *Aspergillus* section *flavi* using MALDI-TOF MS. Noteworthy, this section contains species of both technological and toxigenic interest [34]. The *Aspergillus* section *nigri* is also particularly relevant due to its mycotoxin-producing species and their frequent occurrence in food matrices [33,35]. It is worth noting that this species complex was recently re-examined by Bian et al. [33], and six species were defined within this complex, i.e., *Aspergillus brasiliensis*, *Aspergillus eucalypticola*, *Aspergillus luchuensis*, *Aspergillus niger*, *Aspergillus tubingensis* and *Aspergillus vadensis*. As of today, many species complexes remain to be studied using MALDI-TOF MS. Additionally, the rapid and accurate identification of species within species complex using an updated spectral database could facilitate distinguishing species with diverse incidences and boost the prevention and control of fungal spoilage in food.

In this context, the goal of this study was to expand the VITEK^®^ MS spectral database with food spoilage fungi and evaluate its performance, with a specific emphasis on species differentiation within species complexes.

## 2. Materials and Methods

### 2.1. Fungal Strains

In the present study, the current VITEK^®^ MS V3.4 Knowledge Base was expanded through the addition of 380 yeast and mold strains, which corresponded to 51 genera and 133 species, as detailed in Table 1. Spectra were also acquired on strains belonging to species already present in the previous version of the database. Out of these 133 species, 119 corresponded to new species entries, and 14 corresponded to existing species in the database for which additional spectra were acquired on new strains. Moreover, six mold species within species complexes were reworked, without spectra addition, to optimize their identification accuracy. Species selection was based on their agri-food relevance and prevalence, their ability to colonize various food types and their known mycotoxin production. Strains were obtained from several collections, i.e., American Type Culture Collection (ATCC, Manassas, CO, USA), bioMérieux strain collection (Marcy L’etoile, France), EQUASA industrial strain collection (Plouzané, France), Université de Bretagne Occidentale Culture Collection (UBOCC, Plouzané, France) and the Westerdijk Fungal Biodiversity Institute (Utrecht, The Netherlands). Morphological analysis was performed on all strains to confirm the genus or species. Furthermore, for 330 out of 380 strains, their identification was confirmed using DNA sequencing. One or more genes were sequenced (e.g., ITS region, D1/D2 domain of the 26S rRNA gene, partial ß-tubulin gene, partial elongation factor-1 alpha gene, partial actin gene, glyceraldehyde-3-phosphate dehydrogenase gene).

An external validation was performed to challenge the extended database using 78 strains (Table 2). These strains were obtained from the UBOCC, EQUASA (Plouzané, France) and LUBEM (Plouzané, France) collections. All strains were identified by the DNA sequencing of one or more regions. The list of chosen strains comprised a total of 61 species, including 47 mold species and 14 yeast species. Additionally, 58 species were represented in the extended version of the database, among which 21 represented newly added species, and 3 were extended with additional spectra. The remaining three species were absent from the spectral database.

### 2.2. Strains Inoculation and Cultivation

First, cryopreserved strains were pre-cultured on Sabouraud Dextrose Agar (SDA, bioMérieux, Marcy l’Etoile, France) at 25 °C typically for 2 to 6 d to assess viability and purity. Prior to spectrum acquisition, yeasts were grown at 25 °C for 2 d before spectra acquisition on four different media, i.e., SDA, Malt Extract Agar (MEA), Yeast Glucose Chloramphenicol agar (YGC) and Oxytetracycline Glucose Agar base (OGA). SDA, MEA and YGC were obtained from three different suppliers, i.e., bioMérieux (Marcy l’Etoile, France), Becton Dickinson (BD, Le Pont de Claix, France) and Oxoid—Thermo Fischer Scientific (Dardilly, France), while OGA was obtained from Condalab (Torrejón d’Aedoz, Spain). Molds were cultivated at 25 °C for 2 and 8 d before spectra acquisition with the exception of *Aspergillus restrictus*, *Cladosporium allicinum*, *Penicillium funiculosum*, *Penicillium islandicum* and *Xeromyces bisporus* which were incubated for 8 and 14 d due to their slow growth. Four different media, SDA, MEA, YGC and Potato Dextrose Agar (PDA), from 3 suppliers (bioMérieux, Oxoid and BD) were used for cultivation. The extreme xerophile, *X. bisporus*, was grown in an inhouse agar medium (275 g/L glucose, 275 g/L fructose, 10 g/L malt extract, 2.5 g/L yeast extract, 10 g/L agar, a_w_ 0.84) as described previously [36].

### 2.3. Mold Sample Preparation

After cultivation for 2 and 8 d as mentioned above, mold isolates were subjected to an extraction protocol using the VITEK^®^ MS mold kit (bioMérieux, Marcy l’Etoile, France). Briefly, the mycelium and/or the conidia were sampled on the agar plate surface (approximately 1 cm^2^) using a sterile cotton swab moisturized with API Suspension Medium (bioMérieux, Marcy l’Etoile, France) [18]. The sample was then immersed into a microcentrifuge tube filled with 900 µL of 70% ethanol (bioMérieux, Marcy l’Etoile, France). After vortexing for 5 s and centrifugation for 2 min at 14,000× *g*, the supernatant was discarded, and the pellet was resuspended into 40 µL of 70% formic acid (bioMérieux, Marcy l’Etoile, France). After vortexing for 5 s, 40 µL of acetonitrile was added and vortexed again for 5 s. Finally, a 2 min centrifugation was carried out at 14,000× *g*, and the supernatant was kept for spectra acquisition.

### 2.4. Spectra Acquisition

For spectra acquisition, two distinct protocols were applied for yeast and mold isolates. For yeast isolates, one colony was randomly collected using a loop or the VITEK^®^ PICKME^TM^ (bioMérieux, Marcy l’Etoile, France) and then smeared in duplicate on a target slide (bioMérieux, Marcy l’Etoile, France). Then, 1 µL of 70% formic acid (bioMérieux, Marcy l’Etoile, France) was added directly to each spot and left to dry. For mold isolates, 1 µL of the previously obtained supernatant was transferred in duplicate on the target slide and allowed to dry. Then, for both yeasts and molds, 1 µL of α-cyano-hydroxycinnamic acid matrix solution (CHCA, bioMérieux, Marcy l’Etoile, France) was applied, and the spots were left to dry before MALDI-TOF MS analysis.

Spectra acquisition was performed using the VITEK^®^ MS system (bioMérieux, Marcy l’Etoile, France) equipped with the Launchpad version 2.9.5.6 acquisition software. As described by Girard et al. [28], spectra were acquired in linear positive extraction mode in a mass range from 2000 to 20,000 Da using the “Auto-Quality” option. Each spectrum was generated by the accumulation of 500 laser shots, 100 profiles being acquired from each spot with five shots per profile. Calibration was externally made using fresh cells of *Escherichia coli* ATCC 8739. Two quality control strains, *A. brasiliensis* ATCC 16404 for molds and *Candida glabrata* MYA—3950 for yeasts, were also included for each reagent kit and on each day of spectra acquisition. The Launchpad acquisition software automatically processed raw spectra through smoothing and peak detection procedures [28].

### 2.5. Spectra Quality Control Procedure

Raw spectra were individually controlled for peak resolution, the signal-to-noise ratio and absolute signal intensity. Spectra used to develop the spectral database present typically between 80 and 200 peaks. Good-quality spectra were subsequently transformed into peak lists containing *m*/*z* values and corresponding intensities [28]. A single linkage agglomerative clustering algorithm was used to generate dendrograms for each species, comparative dendrograms with closely related species and dendrograms involving spectra already included in the database when needed. Dendrograms were then analyzed to detect any doubtful strains and confirm dataset consistency. The acceptance criteria were a minimum of 50% similarity and 50 peaks in common between individual spectra for a given mold species, while a minimum similarity of 65% and 50 peaks in common were used for yeast species.

### 2.6. Non-Supervised Analysis of Spectra from Species within Species Complexes

In the case of species complexes in the database, a non-supervised approach was employed as the first step to assess the discriminatory ability of MALDI-TOF MS. The t-Stochastic Neighbor Embedding (SNE) method was used to visualize the distance between spectra in each species complex using Plotly.js V 2.27.0 [37,38]. This non-linear projection technique enables the visualization of high-dimensional data in a lower dimension, typically a two- or three-dimensional map. The high-dimensional data are converted into a matrix of pairwise similarities followed by the application of t-SNE and visualized in a scatterplot [38]. This dimensionality reduction method aims to preserve as much of the significant structure of the initial data while balancing attention between local and global aspects, thereby reducing the tendency for data points to crowd densely in the center of the map [39].

### 2.7. Development of Spectral Database

As previously described by Girard et al. [28], each peak from the peak list was assigned to one of the 1300 bins within the mass range of 3000 to 17,000 Da [40]. Then, a log base scaling of the peak intensities was applied followed by an L1-normalization. For each species, a predictive model was established using the Advanced Spectra Classifier (ASC) algorithm developed by bioMérieux to obtain a specific weight bin matrix. To provide an identification, the new spectra were compared to the bin weight matrix, and the sum of matching bin weights was calculated and then considered as an intermediate score [28]. The resulting specific scores were transformed into multiclass probability estimates using a Gaussian calibration procedure. A decision algorithm was used to retain only significant matches. When only one species was retained, the result was considered as a ‘single choice’. A ‘low discrimination’ result was obtained when more than one species was proposed, while a ‘no identification’ result was obtained either when no significant matches were found or if more than four different species were retained.

### 2.8. Evaluation of Identification Performance by Cross-Validation

A 5-fold cross-validation was used to optimize the VITEK^®^ MS Knowledge Base and to assess how accurately it would perform on independent new spectra. This process was based on the partitioning of the spectral data into five complementary subsets. As described by Girard et al. [28], one round of cross-validation involved a learning phase on four subsets and the validation of the predictive model on the remaining subset. Five rounds of cross-validation were performed by the permutation of the subsets. The estimated identification performance was obtained by combining the results of each round. A ‘correct identification’ was attributed when the same identification results were obtained between the cross-validation and reference identification. A ‘low discrimination’ result was considered correct if the expected identification was included among the matches. A ‘misidentification’ was considered as a discordant identification between the cross-validation and reference identification. A ‘no identification’ result could also occur implying that the spectrum was considered not identified in this case.

### 2.9. Evaluation of Identification Performance by External Validation

The spectral database was challenged using an external dataset of 78 strains. For cultivation, a medium among those cited above was randomly chosen for each strain. Yeasts were incubated for 2 d before spectra acquisition, while mold isolates were analyzed at two randomly selected incubation times ranging from 2 to 8 d. Positive and negative controls were made using the quality control strains and reagents only, respectively. Spectra acquisition was performed in duplicate as described above. The obtained spectra were compared to the constructed spectral database to evaluate the percentage of correct identification for the species claimed in the database and the absence of identification for those not included in the database.

## 3. Results

### 3.1. Performance Estimation by Cross-Validation and Database Validation

#### 3.1.1. Performance Evaluation by Cross-Validation

The database performance for each species was estimated using cross-validation. Overall, 96.55% of the spectra from the VITEK MS fungal knowledge base were correctly identified to the species level, 3.1% were not identified and 0.35% were erroneously identified (discordant status).

Among the 139 species added to the spectral database, 109 yielded an overall correct identification rate of 100% after cross-validation (Table 3). These species also yielded 100% of spectra assigned as a single choice except for six of them, namely *Aspergillus amoenus*, *Aspergillus tabacinus*, *Candida variabilis*, *Penicillium biforme*, *P. funiculosum* and *Penicillium rubens*, which yielded between 2.38% and 12.28% spectra with low discrimination for *A. tabacinus* and *P. biforme*, respectively. Among the remaining species, an overall correct identification percentage above 90% was achieved for 21 species ranging from 90.91% to 98.57% for *Aspergillus jensenii* and *Penicillium macrosporum*, respectively, while for 5 species (i.e., *Aspergillus creber*, *A. restrictus*, *Cladosporium macrocarpum*, *Colletotrichum siamense*, *Hannaella luteola*), the percentage of spectra correctly identified was between 80% and 90%. Finally, the spectra of four species had levels of correct identification below 80%, i.e., *Aspergillus fischeri* (76.92%), *A. luchuensis* (77.27%), *Colletotrichum tropicale* (73.68%) and *Fusarium verticillioides* (73.53%). *A. fischeri* and *A. luchuensis* had a low percentage of discordant and low discriminant spectra. For *A. fischeri,* 15.38% spectra were identified as a species from the same genus, i.e., *A. coreanus*, while 7.69% of spectra yielded low discrimination results with *A. coreanus* as well. Concerning *A. luchuensis*, 18.18% of spectra were only identified as belonging to the *Aspergillus* series *Nigri*. Furthermore, 26.32% and 25% of spectra from *C. tropicale* and *F. verticillioides* were not identified, respectively.

The cross-validation approach is the first method to evaluate performance and highlight possible cross-identifications. To go further in the evaluation of identification performance, an external validation was conducted with strains not included in the database.

#### 3.1.2. Database Validation

The database was challenged using an external dataset. Overall, the external validation performances were the following for the species present in the database: 89.42% spectra were correctly identified, 8.65% were not identified and 1.92% were misidentified. For 62 out of 75 strains for which species were represented in the database, all acquired spectra showed expected results, i.e., a correct identification (Table 4). Species which did not yield satisfactory results were *Arthrographis kalrae*, *A. creber*, *A. jensenii*, *Chrysosporium keratinophilum*, *Engyodontium album*, the *Fusarium solani* complex, *Hortaea werneckii*, *Mucor plumbeus*, *Mucor piriformis*, *Penicillium aurantiogriseum*, *P. biforme* and *Zygotorulaspora mrakii*. Spectra from these species were either unidentified or inappropriately identified. Among those, only six species, i.e., *A. kalrae*, *A. creber*, *C. keratinophilum*, *C. gloeosporioides*, the *F. solani* complex, *M. piriformis*, *P. aurantiogriseum*, had less than 60% correctly identified spectra. Noteworthily, erroneously identified spectra were assigned to the correct genus. Indeed, spectra from *A. creber* were misidentified as *Aspergillus versicolor.*

Concerning the three strains belonging to species that were not part of the database, all acquired spectra for two of them, i.e., *Mucor brunneogriseus* and *Rhodotorula babjevae*, yielded a “no identification” result, while spectra from *Cladosporium snafimbriatum* were identified as *C. allicinum*/*C. macrocarpum* (low discrimination). It is worth mentioning that *C. snafimbriatum*, a newly described species, is a member of the *Cladosporium herbarum* complex and is also closely related to *C. allicinum* and *C. macrocarpum* [41]. 

### 3.2. Performance Evaluation of MALDI-TOF MS for Species Complex Differentiation

The ability of MALDI-TOF MS for discriminating species within five species complexes, i.e., *Aspergillus* series *Nigri*, *Colletotrichum acutatum* complex, *Mucor circinelloides* complex, *Colletotrichum gloeosporioides* complex and *Bisifusarium dimerum* complex, was evaluated using non-supervised (t-SNE) and supervised approaches (cross-validation). All recognized species within these species complexes were analyzed using MALDI-TOF MS with the exception of *Colletotrichum asianum* and *Bisifusarium tonghuanum* that could not be obtained from international culture collections. The spectra from the five species complexes are displayed on t-SNE maps (Figure 1). As shown in Figure 1A and Appendix A, some species from the *Aspergillus* series *nigri*, such as *A. brasiliensis*, *A. tubingensis*, *A. luchuensis* (ex ‘*Aspergillus coreanus*’) and *A. vadensis*, were distinguishable with well-grouped spectra according to their respective species. Spectra from *A. niger* and those from ‘*Aspergillus lacticoffeatus*’ and ‘*Aspergillus foetidus*’ which are now considered as synonyms of *A. niger* were grouped together which is consistent with Bian et al. [33]. Noteworthily, the effect of cultivation time was visible for two species, i.e., ‘*Aspergillus piperis*’ (synonym of *A. luchuensis*) and *A. eucalypticola*. For ‘*Aspergillus piperis*’ (synonym of *A. luchuensis*), spectra obtained after 8 d were grouped on the upper quadrant, one group on the left and one on the right, while the 2-day spectra were on the lower quadrant. The same results were also observed for *A. eucalypticola* for which 2-day spectra were at the bottom of the lower quadrant, whereas 8-day spectra were at the top of the upper quadrant.

As shown in Figure 1B and Appendix A, spectra from the different species of the *Bisifusarium dimerum* complex were also quite well separated. *B. allantoides*, *B. domesticum* and *B. penicillioides* spectra were grouped on the lower quadrant of the t-SNE map, whereas the remaining species were grouped on the upper quadrant (Figure 1B). Spectra from *B. nectrioides* and *B. delphinoides* appeared to be more closely related on the t-SNE map (Figure 1B) which was also confirmed on the spectral similarity dendrogram (Appendix A, similarity = 65%).

As shown in Figure 1C,D, species within each of the *C. acutatum* and *C. gloeosporioides* complexes demonstrated clear intra-complex separations even though they shared a relatively high-level similarity of over 60% in both cases (Appendix A). Concerning the *C. acutatum* complex, spectra for all of the five species analyzed, i.e., *Colletotrichum nymphaeae*, *Colletotrichum lupini*, *Colletotrichum fioriniae*, *Colletotrichum godetiae* and *C. acutatum*, were well clustered and separated for each species (Figure 1C). As for the *C. gloeosporioides* complex, two species could be easily distinguished on the t-SNE map, i.e., *Colletotrichum fructicola* and *C. gloeosporioides* (Figure 1D). Their scatterplots were distant from each other and from all the other scatterplots. The remaining species, namely *Colletotrichum musae*, *C. siamense* and *C. tropicale*, were mostly grouped on the lower left quadrant. The *C. musae* spectra were well clustered, while the *C. siamense* and *C. tropicale* spectra were interspersed. The spectra for *C. siamense* were mostly present between the two clusters of *C. tropicale* spectra.

As shown in Figure 1E and Appendix A, species from the M. circinelloides complex were also well separated, namely Mucor variicolumellatus, Mucor lusitanicus, Mucor ramosissimus, Mucor janssenii, Mucor ctenidius, Mucor velutinosus and Mucor griseocyanus. Two spectra from the latter species were separated from the others. They were both obtained after cultivation for 2 d on MEA (Oxoid), so it was assumed that it was linked to this specific condition. The impact of incubation time was also noticeable for Mucor bainieri, M. circinelloides and Thamnidium anomalum. For instance, M. bainieri spectra at 2 d post incubation were on the left of the right quadrant, while the spectra at 8 d post incubation were on the right of the left quadrant. The same results, but to a much lower extent, were also observed for M. circinelloides and T. anomalum. Indeed, the spectra of each species were grouped together, but part of the spectra obtained after a 2-day incubation were typically separated from those obtained after an 8-day incubation.

Spectra from the different species of the tested species complex were integrated into the bioMérieux spectral database, and identification performances were assessed by cross-validation (Table 3). The *Aspergillus* series Nigri, comprising currently six species, yielded levels of correct identification ranging from 77.27% to 100% with an overall correct identification of 100% for four species, i.e., *A. brasiliensis*, *A. eucaypticola*, *A. niger* including isolates of ‘*A. foetidus*’ and ‘*A. lacticoffeatus*’ and *A. vadensis*. For the *B. dimerum* complex, which includes nine species, a performance of 100% correct identification was reached. Concerning the *C. acutatum* complex, correct identification rates ranged from 90.48% to 100% where four out of five species were found to yield 100% correct identification. Good performances were also achieved for the *C. gloeosporioides* complex with correct identification levels ranging from 73.68% to 100% and spectra from three out of five species yielding 100% correct identification. Finally, for the *M. circinelloides* complex, correct spectra identification ranged from 95% to 100%, and nine of the ten species had a 100% correct identification level.

## 4. Discussion

### 4.1. Performance Estimation by Cross-Validation and Database Validation

In a previous study, Quéro et al. [42] complemented the VITEK^®^ MS database using 136 species encountered in the food and feed industry demonstrating the importance of an updated database for fungal identification. In the present study, the VITEK^®^ MS Knowledge Base was further reinforced with 119 new selected species and 20 species already present in the database for which improvements were made. The overall cross-validation performance was 96.55% with 3.1% unidentified and 0.35% misidentified.

Overall, 97.12% of the species examined under this study (Table 3) had a correct identification rate ranging from 80% to 100%. Additionally, 109 out of 139 species yielded an overall correct identification rate of 100%, accounting for 78.42% of the species examined. However, the correct identification rates of four species fell below 80%, i.e., *A. fischeri* (76.92%), *A. luchuensis* (77.27%), *C. tropicale* (73.68%) and *F. verticillioides* (73.53%). This lower identification performance could be linked to cross-identifications between closely related species in the database. For instance, *A. luchuensis* had a level of discordant spectra of 18.18%, which were identified as the “*Niger* complex” rather than at the species level. *Aspergillus luchuensis* is indeed one of the species of the *Niger* clade. Species within this clade share a high similarity with one another, and it is difficult to distinguish them despite the use of multigenic DNA barcoding [33,35]. For *F. verticillioides*, we have no clear explanation for this result that was also previously observed by Quero et al. [18]. The fact that 25% of spectra yielded no identification during cross-validation may be caused by the high genetic diversity of *F. verticillioides* at the intra-species level and/or the existence of yet-to-be-identified cryptic species [43]. Therefore, the enrichment of the database with spectra of a larger diversity of strains at the population level could improve identification performance for this species.

The cross-validation results provided an estimation of the database performance. An external validation using strains not included in the database was necessary to assess the identification performance. Considering the results for all tested strains, it is promising that 90.48% spectra were correctly assigned as expected. The misidentified spectra were ascribed to either the closely related species from the same genus or from the same species complex or both. For instance, spectra from *C. snafimbriatum* were identified as *C. allicinum* and *C. macrocarpum*, two closely related species within the same species complex [41]. To address this, the database could be expanded in a future version to encompass more species from the *C. herbarum* complex, including *C. snafimbriatum*. Noteworthily, the external and cross-validation results were consistent. *A. creber*, for example, had an overall identification performance of 89.19% during cross-validation with 8.11% spectra showing low discrimination with *Aspergillus versicolor*. The same result was also found during external validation. This issue could be addressed by adding more strains from the species of the *Aspergillus* series *versicolores* to optimize the identification accuracy of the spectral database. Noteworthily, a simplified classification of the series *versicolores* with a lower number of cryptic species was recently proposed by Sklenář et al. [44], leading to the definition of only four species instead of seventeen. As mentioned by Sklenář et al. [44], the use of this classification for spectral database construction may also improve identification accuracy for this series.

### 4.2. Performance Evaluation of MALDI-TOF MS for Species Complex Differentiation

In total, five species complexes were studied using non-supervised (t-SNE) and supervised approaches. The t-SNE method was used as an unsupervised technique to study closely related species and to assess the discriminatory ability of MALDI-TOF MS. As previously seen, this projection enabled us to differentiate species within the same complex despite a high spectral similarity which can be a struggle using only dendrograms. In fact, this non-linear projection technique enables the visualization of high-dimensional data in a lower-dimensional map and discerns specificity that were not perceptible in other arrangements [37,38].

Despite being phylogenetically and genetically close, four out of the six species of the *Aspergillus* series *Nigri* had an overall correct identification of 100% in cross-validation, and two species were above 90%, i.e., *A. luchuensis* (ex ‘*A. coreanus*’) (96.67%) and *A. tubingensis* (91.15%). *A. luchuensis* (ex ‘*Aspergillus piperis*’) was the only one of the sections with a correct identification level under 80%. In Bian et al. [33], the species-level identification of *Aspergillus* section *Nigri* is considered problematic, if not impossible, even using techniques such as DNA sequencing or MALDI-TOF MS. Yet, following the redefined *Aspergillus* series *Nigri* proposed by Bian et al. [33], a cross-validation performance above 90% was obtained for eight out of the nine species. This demonstrates an improvement in intra-specific differentiation within this section using MALDI-TOF MS, which could address the current difficulty of identifying these hazardous mycotoxin producers.

Secondly, *Colletotrichum* complexes, causative agents of anthracnosis, are responsible for food waste, resulting in an important economic impact. They mainly encompass phytopathogenic species that affect a wide variety of hosts causing considerable crop losses. For instance, two prominent complexes, *C. acutatum* and *C. gloeosporioides*, are responsible for fruit crop infections worldwide, leading to massive plant necrosis [45,46]. Because of their similar characteristics, they could be difficult to differentiate. In the present study, MALDI-TOF MS proved to be a good alternative to molecular techniques to discriminate these species within complexes. The cross-validation results for the *C. acutatum* complex showed 90.48% to 100% of spectra being correctly identified with four out of five species having 100% correct identification. The *C. gloeosporioides* species performance levels varied from 73.68% to 100% with three of them achieving 100% correct identification. This identification performance is promising and could significantly enhance disease management strategies and future management outlook [45,46]. The t-SNE method also showed separated scatterplots for each *C. acutatum* complex species. One strain of *C. godetiae* appeared close to the *C. fioriniae* scatterplots which confirmed the results obtained by cross-validation. However, the phylogenetic data did not show a taxonomic misassignment for this distant strain.The apparent vicinity of *C. acutatum*, *C. nymphaeae* and *C. lupini* on one part and *C. fioriniae* and *C. godetiae* on the other part is also consistent with the scientific literature [47].

Thirdly, the *B. dimerum* and *M. circinelloides* complexes were those with the best identification performances, i.e., 100% for all species and 100% correct identification level for every species except *M. velutinosus* (95%), respectively. Interestingly, these complexes are relevant to differentiate for different reasons. Indeed, the *B. dimerum* complex, which belongs to the *Nectriaceae* family, comprises either plant pathogens, species responsible for opportunistic infections and food spoilage but also a species (*B. domesticum*) voluntarily used by cheesemakers to prevent cheese organoleptic defects (stickiness defect) [48,49,50,51]. Noteworthily, the *B. dimerum* complex which reached 100% correct identification to the species level in cross-validation shows clear clusters in two-dimensional projection and a two-group organization. Those results are in agreement with the phylogenetic analysis conducted by Savary et al. [50]. The result of the present study also confirmed the proximity of *B. nectrioides* and *B. delphinoides*.

The *M. circinelloides* species complex includes saprophytic species responsible for food spoilage that are also known as opportunistic pathogens responsible for mucormycosis in immunocompromised patients [52,53]. Different studies have already demonstrated clear differences in virulence [54] and antifungal susceptibilities [55] among species (or forms formerly) within the complex. Given these concerning issues and the identification performance achieved in this study, MALDI-TOF MS can be a powerful asset for discriminating these species that may have varying ecologies and virulence levels. Among this complex, some species were rather well separated on the t-SNE map, whereas some species were not, and it seems to be linked to growth media and incubation time. Nevertheless, it did not impact identification performances during cross-validation. Similar results have been reported by Quéro et al. [18] for other species, i.e., *A. flavus*, *Aureobasidium pullulans* and *P. expansum*. These peculiar cases are important to keep in the VITEK^®^ MS database because it adds spectral diversity and thus allows us to build a robust database.

## 5. Conclusions

In the present study, the existing VITEK MS database was extended with food-relevant fungal species as well as species belonging to species complexes. It appeared that MALDI-TOF MS was a powerful tool to accurately identify these fungal species as well as to discriminate species within species complexes. These results emphasize the importance of continuously enhancing the database by incorporating relevant species and species complexes and taking into account the continuous evolution and progression of fungal taxonomy.

## Figures and Tables

**Figure 1 jof-10-00456-f001:**
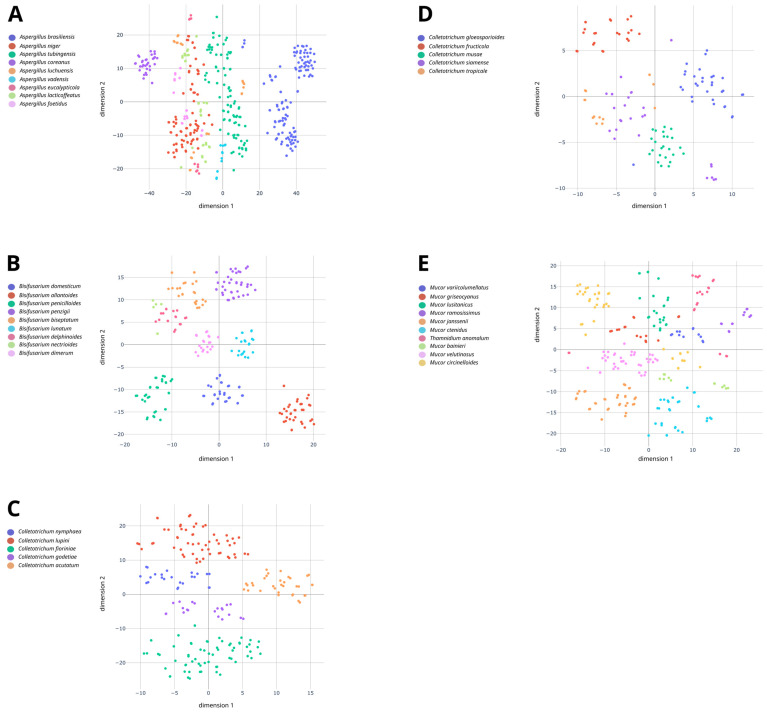
A two-dimensional t-SNE map displaying the spectra from the *Aspergillus* series *nigri* (**A**), the *Bisifusarium dimerum* complex (**B**), the *Colletotrichum acutatum* complex (**C**), the *Colletotrichum gloeosporioides* complex (**D**) and the *Mucor circinelloides* complex (**E**) obtained through MALDI-TOF MS. Spectra are colored according to the respective species to which they belong.

**Table 1 jof-10-00456-t001:** The species and strain numbers for each species used to expand the database.

Species	Strain Number *
*Alternaria brassicicola*	UBOCC-A-101046, UBOCC-A-101047
*Alternaria infectoria*	EQUASA 414, EQUASA 415
*Arthrographis kalrae*	API 2104239, API 2104240, API 2104241, API 2104242
*Aspergillus amoenus*	EQUASA 1271, EQUASA 1423, EQUASA 1492, EQUASA 1502
*Aspergillus cibarius*	EQUASA 218, EQUASA 610, EQUASA 1328
*Aspergillus clavatus*	EQUASA 749, EQUASA 822
*Aspergillus creber*	EQUASA 1424, EQUASA 1425, EQUASA 1491, EQUASA 1499, EQUASA 1504
*Aspergillus domesticus*	UBOCC-A-115038, UBOCC-A-115040
*Aspergillus fischeri*	CBS 125813, CBS 483.65
*Aspergillus hiratsukae*	EQUASA 854, EQUASA 1132, EQUASA 1133, EQUASA 1435
*Aspergillus intermedius* **	CBS 108.55, CBS 523.65 ^NT^, CBS 117329, CBS 116.62
*Aspergillus jensenii*	EQUASA 956, EQUASA 1262, EQUASA 1266, EQUASA 1489
*Aspergillus penicilloides*	CBS 234.65, CBS 130294
*Aspergillus quadricinctus*	CBS 135.52 ^T^, CBS 128010
*Aspergillus restrictus*	UBOCC-A-101080, CBS 541.65 ^T^
*Aspergillus sojae*	CBS 134.52, CBS 100928 ^NT^
*Aspergillus tabacinus*	EQUASA 1018, EQUASA 1427, EQUASA 1488, EQUASA 1500
*Berkeleyomyces basicola*	EQUASA 1024, EQUASA 1088, EQUASA 1089, UBOCC-A-101281
*Candida carpophila*	CBS 5256 ^T^, CBS 5257
*Candida deformans*	EQUASA 711, EQUASA 1313, EQUASA 1314, EQUASA 1320
*Candida pseudoglaebosa*	CBS 6715 ^T^, UBOCC-A-214189
*Candida saitoana*	CBS 6729, CBS 940 ^T^
*Candida variabilis*	EQUASA 816, EQUASA 817, EQUASA 1120, EQUASA 1394
*Candida versatilis*	CBS 1752 ^T^
*Chrysonilia sitophila*	UBOCC-A-101030, UBOCC-A-111120, UBOCC-A-111121, UBOCC-A-111122
*Chrysosporium keratinophilum*	API 2104246, API 2104247, API 2104248, API 2104249
*Cladosporium allicinum*	EQUASA 453, EQUASA 1406, EQUASA 1411
*Cladosporium bruhnei*	EQUASA 239, EQUASA 406, CBS 134.31, CBS 110024
*Cladosporium macrocarpum*	EQUASA 70, EQUASA 91
*Colletotrichum karsti*	UBOCC-A-116037, CBS 132134 ^NT^
*Cunninghamella elegans* **	API 2104252
*Cystobasidium minitum* **	UBOCC-A-214063, UBOCC-A-214082
*Engyodontium album*	EQUASA 474, EQUASA 753, EQUASA 791, EQUASA 1066
*Eupenicillium lapidosum*	EQUASA 1111, EQUASA 1173, EQUASA 1181, EQUASA 1348
*Exophiala bergeri*	CBS 351.52, CBS 353.52 ^T^, CBS 102241, CBS 109786, CBS 111662, CBS 119094, CBS 119099
*Exophiala lecanii-corni*	CBS 232.39 ^T^, API 2101095, API 2101096, API 2101100, API 2101101, API 2101102
*Exophiala oligosperma*	CBS 725.88 ^T^, API 2101105
*Exophiala pisciphila*	EQUASA 373, EQUASA 375
*Exophiala psychrophila*	EQUASA 397, CBS 191.87 ^T^
*Filobasidium magnum*	UBOCC-A-214029, UBOCC-A-214192
*Fusarium verticillioides* **	CBS 734.97
*Geosmithia swiftii* **	CBS 116927, CBS 110774, CBS 158.67, CBS 296.48 ^NT^
*Hannaella luteola*	API 9312069, API 2102042
*Helicostylum elegans*	EQUASA 401.02, EQUASA 402.04, EQUASA 302.01, EQUASA 408.01
*Hortaea werneckii*	EQUASA 88, EQUASA 449, UBOCC-A-201189, UBOCC-A-208029
*Hyphopichia pseudoburtonii*	CBS 5510, CBS 2455 ^T^, EQUASA 1417
*Isomucor fuscus*	CBS 254.48 ^T^, UBOCC-A-109168, UBOCC-A-109169, CBS 230.29, UBOCC-A-109167
*Kazachstania exigua*	CBS 6440, CBS 135 ^NT^
*Kazachstania unispora*	CBS 398 ^T^, UBOCC-A-220043, UBOCC-A-223012, UBOCC-A-223014
*Lachancea kluyveri*	CBS 3082 ^T^, UBOCC-A-201045
*Linnemannia hyalina*	EQUASA 281, EQUASA 282
*Linnemannia zychae*	EQUASA 270, EQUASA 565
*Microdochium nivale* **	UBOCC-A-102041, UBOCC-A-105025, UBOCC-A-113085, UBOCC-A-113088
*Mucor mucedo*	UBOCC-A-109064, CBS 640.67 ^NT^, CBS 887.71
*Mucor piriformis*	EQUASA 582, CBS 169.25 ^NT^
*Nigrospora oryzae*	CBS 382.50, CBS 384.69
*Nigrospora sphaerica*	EQUASA 257, EQUASA 685
*Paraphyton cookei*	API 2104256, API 2104258
*Penicillium biforme*	UBOCC-A-110150, UBOCC-A-112050, UBOCC-A-112051, UBOCC-A-112052, UBOCC-A-112053
*Penicillium charlesii*	CBS 304.48 ^T^
*Penicillium corylophilum* **	EQUASA 86
*Penicillium fellutanum*	CBS 229.81 ^NT^, UBOCC-A-123037
*Penicillium funiculosum*	UBOCC-A-101419, UBOCC-A-112140
*Penicillium glandicola*	CBS 498.75 ^ET^, UBOCC-A-110041
*Penicillium griseofulvum*	UBOCC-A-101424, UBOCC-A-109220
*Penicillium islandicum*	UBOCC-A-101425, CBS 394.50, CBS 165.81
*Penicillium janczewskii*	UBOCC-A-111140, UBOCC-A-113046
*Penicillium janthinellum*	UBOCC-A-101427, UBOCC-A-111189
*Penicillium macrosporum* **	CBS 350.72, CBS 118884, CBS 130.89, CBS 317.63 ^T^
*Penicillium olsonii*	UBOCC-A-117001, UBOCC-A-117002, UBOCC-A-118059, UBOCC-A-118158
*Penicillium purpurogenum*	CBS 128132, CBS 128133, CBS 184.27
*Penicillium rubens*	EQUASA 869, EQUASA 954, EQUASA 955, EQUASA 1265, EQUASA 1268, EQUASA 1490, EQUASA 1509
*Penicillium rugulosum*	EQUASA 936, EQUASA 1506, UBOCC-A-111174, UBOCC-A-111181, UBOCC-A-111190
*Penicillium ubiquetum*	EQUASA 125, EQUASA 129
*Phoma pinodella*	UBOCC-A-116004, CBS 531.66, CBS 133.92, CBS 123522, CBS 403.65
*Pichia occidentalis*	CBS 10322, CBS 6399
*Rhinocladiella similis*	EQUASA 529, EQUASA 942
*Rhizomucor pusillus*	UBOCC-A-101365, UBOCC-A-111202, API 2104260, API 2104264
*Saccharomyces cariocanus*	UBOCC-A-220015, UBOCC-A-220031, UBOCC-A-220045
*Saccharomyces uvarum*	CBS 377, CBS 395 ^T^, UBOCC-A-201049
*Saccharomycopsis fibuligera*	UBOCC-A-212006, UBOCC-A-212010, EQUASA 1082, EQUASA 823
*Schizophillum commune*	API 2104268, API 2104269
*Scopulariopsis asperula* **	UBOCC-A-101272, UBOCC-A-108119, UBOCC-A-110145, UBOCC-A-113016
*Scopulariopsis candida*	UBOCC-A-108117, UBOCC-A-110144, UBOCC-A-113025
*Scopulariopsis flava*	UBOCC-A-108118, UBOCC-A-113028
*Sporobolomyces roseus*	UBOCC-A-214093, UBOCC-A-214118, CBS 486 ^LT^
*Stachybotrys chartarum*	API 2104273, API 2104274, API 2104275, API 2104276
*Starmerella etchellsii*	CBS 1750 ^T^, CBS 1751
*Syncephalastrum racemosum*	API 2104281, API 2104282
*Thamnidium elegans*	CBS 642.69, CBS 341.55
*Trichoderma atroviride*	UBOCC-A-101288
*Trichoderma harzianum* **	UBOCC-A-118023, CBS 226.95 ^NT^, UBOCC-A-117301
*Trichoderma viride* **	UBOCC-A-101292
*Trigonopsis californica*	CBS 5383, CBS 5654
*Verticillium albo-atrum*	EQUASA 1143, UBOCC-A-101307
*Verticillium dahliae*	UBOCC-A-101312, UBOCC-A-101313, UBOCC-A-101314, UBOCC-A-101317
*Verticillium nonalfalfae*	EQUASA 203, EQUASA 589, EQUASA 590, UBOCC-A-112135
*Wallemia muriae*	CBS 116628 ^NT^, CBS 110619, CBS 110624
*Xeromyces bisporus*	CBS 469.59, CBS 347.94, CBS 236.71
*Zygotorulaspora florentina*	CBS 748, CBS 6703, CBS 6761
*Zygotorulaspora mrakii*	UBOCC-A-220020, UBOCC-A-220022, UBOCC-A-220023, UBOCC-A-220024, UBOCC-A-220025
*Aspergillus* series *Nigri*	
*Aspergillus brasiliensis* **	ATCC 16404
*Aspergillus luchuensis* (*ex Aspergillus coreanus*)	CBS 119384, EQUASA 756, EQUASA 1073, EQUASA 1170
*Aspergillus eucalypticola*	CBS 122712 ^HT^
*Aspergillus niger* (ex *Aspergillus foetidus*) ***	CBS 114.49, CBS 121.28 ^NT^
*Aspergillus niger* (ex *Aspergillus lacticoffeatus*) ***	CBS 101884, CBS 101885, CBS 101886
*Aspergillus luchuensis (ex Aspergillus piperis*)	CBS 112811, CBS 113.52, CBS 113.33
*Aspergillus niger* ***	API 1006067, API 1006068, API 1105141, API 1212008, UBOCC-A-101072, UBOCC-A-101075, CBS 554.65, UBOCC-A-112064, UBOCC-A-112068, UBOCC-A-112080, UBOCC-A-112082
*Aspergillus tubingensis* **	CBS 115656 ^HT^, CBS 115657, CBS 132411, CBS 563.65, CBS 115574
*Aspergillus vadensis*	CBS 113226, CBS 113365
*Bisifusarium dimerum* complex	
*Bisifusarium allantoides*	UBOCC-A-120035, UBOCC-A-120036, UBOCC-A-120037
*Bisifusarium biseptatum*	CBS 110138, CBS 110306, CBS 110144
*Bisifusarium delphinoides*	CBS 115321, CBS 101047
*Bisifusarium dimerum* ***	SA132479, SA131363, SA131166
*Bisifusarium domesticum* ***	UBOCC-A-109095, UBOCC-A-113010, CBS 244.82
*Bisifusarium lunatum*	UBOCC-A-118038, CBS 632.76 ^NT^
*Bisifusarium nectrioides*	CBS 176.31 ^LT^
*Bisifusarium penicillioides*	UBOCC-A-120021 ^T^, UBOCC-A-120034, UBOCC-A-120054
*Bisifusarium penzigii*	CBS 317.34 ^HT^, EQUASA 1440, EQUASA 1441, EQUASA 1442
*Colletotrichum acutatum* complex	
*Colletotrichum acutatum*	UBOCC-A-117265, CBS 126505, CBS 129952, CBS 129914
*Colletotrichum fioriniae*	UBOCC-A-116032, UBOCC-A-117425, CBS 128517 ^T^, UBOCC-A-116034, UBOCC-A-116033, UBOCC-A-121023, UBOCC-A-103034
*Colletotrichum godetiae*	CBS 133.44 ^T^, UBOCC-A-121017, UBOCC-A-121021, UBOCC-A-115012
*Colletotrichum lupini*	UBOCC-A-118145, UBOCC-A-118146, UBOCC-A-118147, CBS 109221 ^HT^, CBS 109225
*Colletotrichum nymphaeae*	UBOCC-A-117287, CBS 515.78 ^ET^
*Colletotrichum gloeosporioides* complex	
*Colletotrichum fructicola*	UBOCC-A-118064, UBOCC-A-118065
*Colletotrichum gloeosporioides*	UBOCC-A-116039, UBOCC-A-116038, UBOCC-A-116036
*Colletotrichum musae*	UBOCC-A-121003, UBOCC-A-121004
*Colletotrichum siamense*	UBOCC-A-121006, UBOCC-A-121020, CBS 125379
*Colletotrichum tropicale*	UBOCC-A-121005, CBS 124949 ^HT^, CBS 125389
*Mucor circinelloides* complex	
*Mucor bainieri*	CBS 293.63 ^IT^
*Mucor circinelloides* ***	UBOCC-A-109182, UBOCC-A-109183, CBS 195.68, UBOCC-A-109192
*Mucor ctenidius*	CBS 433.87, CBS 696.76, CBS 293.66
*Mucor griseocyanus*	CBS 116.08, CBS 223.56
*Mucor janssenii*	CBS 232.29, CBS 185.68, CBS 205.68
*Mucor lusitanicus*	CBS 633.65, CBS 851.71, CBS 242.33
*Mucor ramosissimus*	CBS 135.65 ^NT^
*Mucor variicolumellatus*	CBS 236.35 ^HT^
*Mucor velutinosus* **	EQUASA 1551
*Thamnidium anomalum*	CBS 697.76, CBS 243.57 ^T^

* API, bioMérieux culture collection strains; ATCC, American Type Culture Collection; CBS, Westerdijk Fungal Biodiversity Institute culture collection; EQUASA, Etude En Qualité Alimentaire culture collection; UBOCC, Université de Bretagne Occidentale Culture Collection; SA, external site development collection. ** Species already present in the database for which additional strains were integrated. *** Species from species complexes already present in the database and reviewed for the VITEK^®^ MS Knowledge Base Version 3.4. An underlined strain number indicates that strain identification was also confirmed by DNA sequencing. ^T^ Type strain ^HT^ Holotype ^ET^ Epitype ^LT^ Lectotype ^NT^ Neotype ^IT^ Isotype.

**Table 2 jof-10-00456-t002:** Species and strain numbers used for external database validation.

Species	Strain Number
*Alternaria brassicicola*	M1-0046
*Arthrographis kalrae* *	API 2104244
*Aspergillus amoenus* *	EQUASA 1261
*Aspergillus cibarius* *	EQUASA 610
*Aspergillus creber* *	EQUASA 491, EQUASA 1169
*Aspergillus hiratsukae* *	EQUASA 1436
*Aspergillus jensenii* *	EQUASA 2677
*Aspergillus tubingensis* *	M1-0085
*Aureobasidium pullulans*	L1-0011
*Bisifusarium biseptatum / penzigii* *	EQUASA 1442
*Botrytis cinerea*	M1-0123, M2-0036
*Candida famata*	L1-0009
*Candida guilliermondii*	L1-0006
*Candida hellenica*	L1-0022
*Candida pulcherrima*	L2-0005
*Chrysonilia sitophila* *	UBOCC-A-111123, UBOCC-A-111124
*Chrysosporium keratinophilum* *	API 2104251
*Cladosporium cladosporioides* complex	M1-0045, M1-0126, M2-0041
*Cladosporium snafimbriatum* **	M2-0010
*Cladosporium oxysporum*	M1-0011
*Cladosporium ramotenellum*	M1-0014
*Colletotrichum lupini* *	UBOCC-A-118080, UBOCC-A-118081
*Engyodontium album* *	EQUASA 473
*Eupenicillium lapidosum* *	EQUASA 1446
*Exophiala dermatitidis*	L1-0023
*Fusarium proliferatum*	M1-0077, M1-0116
*Fusarium sambucinum*	M1-0137
*Fusarium solani* complex	M1-0061
*Geotrichum candidum*	M1-0054
*Hortaea werneckii* *	EQUASA 680, EQUASA 1364, EQUASA 1365
*Kloeckera apiculata*	L1-0015
*Microdochium nivale* *	UBOCC-A-102027
*Mucor brunneogriseus* **	M1-0063
*Mucor circinelloides*	M1-0152
*Mucor plumbeus*	M1-0139, M1-0153
*Mucor piriformis* *	M2-0003
*Papiliotrema laurentii*	L1-0007
*Penicillium adametzioides*	M1-0020
*Penicillium aurantiogriseum*	M1-0150
*Penicillium aurantiogriseum var. polonicum*	M1-0001
*Penicillium biforme* *	UBOCC-A-112057, UBOCC-A-112058, UBOCC-A-112059
*Penicillium brevicompactum*	M1-0025
*Penicillium citrinum*	M1-0049
*Penicillium crustosum*	M1-0134, M1-0149
*Penicillium digitatum*	M2-0006
*Penicillium expansum*	M1-0079
*Penicillium glabrum*	M2-0033
*Penicillium italicum*	M1-0144
*Penicillium olsonii* *	UBOCC-A-118177, UBOCC-A-118178
*Penicillium paneum*	M1-0109
*Penicillium rubens* *	EQUASA 1459, EQUASA 1448
*Penicillium solitum*	M1-0047
*Phoma glomerata*	M1-0108, CBS 318.90
*Rhizopus stolonifer*	M1-0098
*Rhodotorula babjevae* **	L1-0002
*Rhodotorula mucilaginosa*	L1-0018
*Sporobolomyces roseus* *	UBOCC-A-208018
*Trichoderma harzianum* *	M1-0140
*Trichoderma viride/ghanense* *	M1-0081
*Verticillium nonalfalfae* *	EQUASA 526
*Zygotorulaspora mrakii* *	UBOCC-A-220032, UBOCC-A-220040

* Species added to the spectral database in the present study. ** Species not included in the database.

**Table 3 jof-10-00456-t003:** The performance evaluation of the database by cross-validation.

Species	Overall Correct (%) ^(1)^	Single Choice (%)	Low Discrimination (%)	No Identification (%)	Discordant (%)
*Alternaria brassicicola*	100	100	0	0	0
*Alternaria infectoria*	100	100	0	0	0
*Arthrographis kalrae*	100	100	0	0	0
*Aspergillus amoenus*	100	89.47	10.53	0	0
*Aspergillus cibarius*	100	100	0	0	0
*Aspergillus clavatus*	98.53	98.53	0	1.47	0
*Aspergillus creber*	89.19	81.08	8.11	10.81	0
*Aspergillus domesticus*	100	100	0	0	0
*Aspergillus fischeri*	76.92	69.23	7.69	7.69	15.38
*Aspergillus hiratsukae*	98.36	98.36	0	1.64	0
*Aspergillus intermedius* *	100	100	0	0	0
*Aspergillus jensenii*	90.91	86.36	4.55	4.55	4.55
*Aspergillus penicilloides*	100	100	0	0	0
*Aspergillus quadricinctus*	92.86	92.86	0	7.14	0
*Aspergillus restrictus*	87.5	87.5	0	3.13	9.38
*Aspergillus sojae*	100	100	0	0	0
*Aspergillus tabacinus*	100	97.62	2.38	0	0
*Berkeleyomyces basicola*	100	100	0	0	0
*Candida carpophila*	91.67	91.67	0	8.33	0
*Candida deformans*	100	100	0	0	0
*Candida pseudoglaebosa*	100	100	0	0	0
*Candida saitoana*	100	100	0	0	0
*Candida variabilis*	100	93.75	6.25	0	0
*Candida versatilis*	100	100	0	0	0
*Chrysonilia sitophila*	100	100	0	0	0
*Chrysosporium keratinophilum*	100	100	0	0	0
*Cladosporium allicinum*	100	100	0	0	0
*Cladosporium bruhnei*	100	100	0	0	0
*Cladosporium macrocarpum*	85.71	42.86	42.86	7.14	7.14
*Colletotrichum karsti*	90	90	0	10	0
*Cunninghamella elegans* *	100	100	0	0	0
*Cystobasidium minitum* *	97.44	87.18	10.26	2.56	0
*Engyodontium album*	94.12	94.12	0	5.88	0
*Eupenicillium lapidosum*	100	100	0	0	0
*Exophiala bergeri*	96.55	96.55	0	1.72	1.72
*Exophiala lecanii-corni*	100	100	0	0	0
*Exophiala oligosperma*	100	100	0	0	0
*Exophiala pisciphila*	100	100	0	0	0
*Exophiala psychrophila*	100	100	0	0	0
*Filobasidium magnum*	100	100	0	0	0
*Fusarium verticillioides* *	73.53	73.53	0	18.18	0
*Geosmithia swiftii* *	100	100	0	0	0
*Hannaella luteola*	81.82	81.82	0	18.18	0
*Helicostylum elegans*	100	100	0	0	0
*Hortaea werneckii*	100	100	0	0	0
*Hyphopichia pseudoburtonii*	100	100	0	0	0
*Isomucor fuscus*	100	100	0	0	0
*Kazachstania exigua*	100	100	0	0	0
*Kazachstania unispora*	100	100	0	0	0
*Lachancea kluyveri*	100	100	0	0	0
*Linnemannia hyalina*	100	100	0	0	0
*Linnemannia zychae*	100	100	0	0	0
*Microdochium nivale* *	100	100	0	0	0
*Mucor mucedo*	100	100	0	0	0
*Mucor piriformis*	95	95	0	5	0
*Nigrospora oryzae*	100	100	0	0	0
*Nigrospora sphaerica*	100	100	0	0	0
*Paraphyton cookei*	100	100	0	0	0
*Penicillium biforme*	100	87.72	12.28	0	0
*Penicillium charlesii*	90	80	10	10	0
*Penicillium corylophilum* *	100	100	0	0	0
*Penicillium fellutanum*	100	100	0	0	0
*Penicillium funiculosum*	100	93.75	6.25	0	0
*Penicillium glandicola*	100	100	0	0	0
*Penicillium griseofulvum*	100	100	0	0	0
*Penicillium islandicum*	100	100	0	0	0
*Penicillium janczewskii*	100	100	0	0	0
*Penicillium janthinellum*	100	100	0	0	0
*Penicillium macrosporum* *	98.57	98.57	0	1.43	0
*Penicillium olsonii*	100	100	0	0	0
*Penicillium purpurogenum*	93.33	93.33	0	3.33	3.33
*Penicillium rubens*	100	96.43	3.57	0	0
*Penicillium rugulosum*	100	100	0	0	0
*Penicillium ubiquetum*	95.65	95.65	0	4.35	0
*Phoma pinodella*	100	100	0	0	0
*Pichia occidentalis*	100	100	0	0	0
*Rhinocladiella similis*	92.86	92.86	0	7.14	0
*Rhizomucor pusillus*	100	100	0	0	0
*Saccharomyces cariocanus*	100	100	0	0	0
*Saccharomyces uvarum*	100	100	0	0	0
*Saccharomycopsis fibuligera*	100	100	0	0	0
*Schizophillum commune*	100	100	0	0	0
*Scopulariopsis asperula* *	100	100	0	0	0
*Scopulariopsis candida*	100	100	0	0	0
*Scopulariopsis flava*	100	100	0	0	0
*Sporobolomyces roseus*	100	100	0	0	0
*Stachybotrys chartarum*	100	100	0	0	0
*Starmerella etchellsii*	100	100	0	0	0
*Syncephalastrum racemosum*	100	100	0	0	0
*Thamnidium elegans*	100	100	0	0	0
*Trichoderma atroviride*	100	0	100	0	0
*Trichoderma harzianum* *	100	100	0	0	0
*Trichoderma viride* *	100	0	100	0	0
*Trigonopsis californica*	100	100	0	0	0
*Verticillium albo-atrum*	100	0	100	0	0
*Verticillium dahliae*	97.37	97.37	0	2.63	0
*Verticillium nonalfalfae*	100	0	100	0	0
*Wallemia muriae*	100	100	0	0	0
*Xeromyces bisporus*	96.67	96.67	0	3.33	0
*Zygotorulaspora florentina*	100	100	0	0	0
*Zygotorulaspora mrakii*	100	100	0	0	0
*Aspergillus nigri* section					
*Aspergillus brasiliensis* *	100	100	0	0	0
*Aspergillus luchuensis (ex Aspergillus coreanus*)	96.97	96.97	0	3.03	0
*Aspergillus eucalypticola*	100	0	100	0	0
*Aspergillus niger* (ex *Aspergillus foetidus*) **	100	100	0	0	0
*Aspergillus niger* (ex *Aspergillus lacticoffeatus*) **	100	100	0	0	0
*Aspergillus luchuensis (ex Aspergillus piperis*)	77.27	0	77.27	4.55	18.18
*Aspergillus niger* **	100	100	0	0	0
*Aspergillus tubingensis* *	91.15	91.15	0	4.42	4.42
*Aspergillus vadensis*	100	100	0	0	0
*Bisifusarium dimerum* complex					
*Bisifusarium allantoides*	100	100	0	0	0
*Bisifusarium biseptatum*	100	100	0	0	0
*Bisifusarium delphinoides*	100	0	100	0	0
*Bisifusarium dimerum* **	100	100	0	0	0
*Bisifusarium domesticum* **	100	100	0	0	0
*Bisifusarium lunatum*	100	100	0	0	0
*Bisifusarium nectrioides*	100	0	100	0	0
*Bisifusarium penicillioides*	100	100	0	0	0
*Bisifusarium penzigii*	100	100	0	0	0
*Colletotrichum acutatum* complex					
*Colletotrichum acutatum*	100	100	0	0	0
*Colletotrichum fioriniae*	100	100	0	0	0
*Colletotrichum godetiae*	90.48	90.48	0	9.52	0
*Colletotrichum lupini*	100	100	0	0	0
*Colletotrichum nymphaeae*	100	100	0	0	0
*Colletotrichum gloeosporioides* complex					
*Colletotrichum fructicola*	100	100	0	0	0
*Colletotrichum gloeosporioides*	100	100	0	0	0
*Colletotrichum musae*	100	100	0	0	0
*Colletotrichum siamense*	82.14	0	82.14	17.86	0
*Colletotrichum tropicale*	73.68	0	73.68	26.32	0
*Mucor circinelloides* complex					
*Mucor bainieri*	100	0	100	0	0
*Mucor circinelloides* **	100	100	0	0	0
*Mucor ctenidius*	100	100	0	0	0
*Mucor griseocyanus*	100	100	0	0	0
*Mucor janssenii*	100	100	0	0	0
*Mucor lusitanicus*	100	0	100	0	0
*Mucor ramosissimus*	100	0	100	0	0
*Mucor variicolumellatus*	100	0	100	0	0
*Mucor velutinosus* *	95	95	0	5	0
*Thamnidium anomalum*	100	100	0	0	0

* Species already present in the database for which additional strains were integrated. ** Species from species complexes already present in the database. ^(1)^ Single choice stands for spectra identified to the correct species, low discrimination corresponds to spectra which matched with different species including the correct one and the overall correct percentage results of the addition of single choice and low discrimination percentages.

**Table 4 jof-10-00456-t004:** Performance evaluation of the database by external validation.

	Species (Number of Strains)	Number of Spectra	Number of Correct Identification	Number of No Identification	Number of Misidentification
Species present in the database					
*Alternaria brassicicola*	1	2	2		
*Arthrographis kalrae*	1	4	2	2	
*Aspergillus amoenus*	1	4	4		
*Aspergillus cibarius*	1	4	4		
*Aspergillus creber*	2	12	6	2	4 (*Aspergillus versicolor*)
*Aspergillus hiratsukae*	2	6	6		
*Aspergillus jensenii*	1	3	2	1	
*Aspergillus tubingensis*	1	2	2		
*Aureobasidium pullulans*	1	2	2		
*Bisifusarium biseptatum / penzigii*	1	4	4		
*Botrytis cinerea*	2	4	4		
*Candida famata*	1	2	2		
*Candida guilliermondii*	1	2	2		
*Candida hellenica*	1	2	2		
*Candida pulcherrima*	1	2	2		
*Chrysonilia sitophila*	2	10	10		
*Chrysosporium keratinophilum*	1	4	2	2	
*Cladosporium cladosporioides* complex	3	6	6		
*Cladosporium oxysporum*	1	2	2		
*Cladosporium ramotenellum*	1	2	2		
*Colletotrichum lupini*	2	8	8		
*Engyodontium album*	1	6	5	1	
*Eupenicillium lapidosum*	1	4	4		
*Exophiala dermatitidis*	1	2	2		
*Fusarium proliferatum*	2	4	4		
*Fusarium sambucinum*	1	2	2		
*Fusarium solani* complex	1	2	1	1	
*Geotrichum candidum*	1	2	2		
*Hortaea werneckii*	3	6	5	1	
*Kloeckera apiculata*	1	2	2		
*Microdochium nivale*	1	5	5		
*Mucor circinelloides*	1	2	2		
*Mucor plumbeus*	2	4	3	1	
*Mucor piriformis*	1	2	2		
*Papiliotrema laurentii*	1	2	1	1	
*Penicillium adametzioides*	1	2	2		
*Penicillium aurantiogriseum*	1	2		2	
*Penicillium aurantiogriseum var. polonicum*	1	2	2		
*Penicillium biforme*	3	12	10	2	
*Penicillium brevicompactum*	1	2	2		
*Penicillium citrinum*	1	2	2		
*Penicillium crustosum*	2	4	4		
*Penicillium digitatum*	1	2	2		
*Penicillium expansum*	1	2	2		
*Penicillium glabrum*	1	2	2		
*Penicillium italicum*	1	2	2		
*Penicillium olsonii*	2	8	8		
*Penicillium paneum*	1	2	2		
*Penicillium rubens*	2	8	8		
*Penicillium solitum*	1	2	2		
*Phoma glomerata*	2	6	6		
*Rhizopus stolonifer*	1	2	2		
*Rhodotorula mucilaginosa*	1	2	2		
*Sporobolomyces roseus*	1	2	2		
*Trichoderma harzianum*	1	2	2		
*Trichoderma viride/ghanense*	1	2	2		
*Verticillium nonalfalfae*	1	2	2		
*Zygotorulaspora mrakii*	1	4	2	2	
Species absent in the database					
*Cladosporium snafimbriatum*	1	2			2 (*Cladosporium allicinum* and *Cladosporium macrocarpum*)
*Mucor brunneogriseus*	1	2		2	
*Rhodotorula babjevae*	1	2		2	

## Data Availability

The original contributions presented in this study are included in the article/Appendix A; further inquiries can be directed to the corresponding author.

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
