# Peer review of "Identification of Food Spoilage Fungi Using MALDI-TOF MS: Spectral Database Development and Application to Species Complex"

_jof, 2024, doi:10.3390/jof10070456_

Round 1

Reviewer 1 Report

The manuscript by Rolland et al. presented the development & application of spectral database for the identification of food spoilage fungi at the species level within species complex via MALDI-TOF MS. This is of significance in the food industry and is promising to use in routine food safety control. Although this paper is an extended and updated work on the basis of the previous publication by the same group, Quero et al. (Food Microbiology 2019, 81, 76-88), it is suggested that the fungal strains and cultivation conditions for external validation should be improved or at least kept consistent with the level described in previous publication Quero et al.( 2019): “Each strain was cultivated under two different conditions. For each condition, an incubation time varying between 2 and 8 days…”.

Minor issues:

11.    Vocabulary:

-        P2 Line 58-59, in the sentence  of “…rapid and accurate identification of fungi to the species level is crucial to ensure food safety…” :

to> at

-        P2 Line 70-71, “…requiring special skills and knowledge to be applicable in routine in an industrial context>”requiring special skills and knowledge to be applicable in routine examination at the industrial setting”.

-         P2 Line 94, “…of technological interest and toxigenic ones” > “…of technological and toxigenic interests”

-        P3 Line 100-103, in the sentence of “…rapid and accurate identification of species within species complex using an updated spectral database could help distinguish species with diverse incidences and optimize prevention and control of fungal spoilage in food.”

            help distinguish> facilitate distinguishing; 

          optimize prevention > boost the prevention

-        P3 Line 104, the aim > the goal

-        P9 Line 266, ranging between 2 and 8 days >ranging from 2 to 8 days

         Line 267, “Positive and negative controls were made using respectively only reagents…”   respectively only reagents > the corresponding reagents only

     Line 271, “…for those not in the database.> … for those not included in the database

-       P13 Line 318, “were either not or incorrectly identified> were either unidentified or inappropriately identified.

-       P13 Line 323, … not included in the database >  excluded in the database

-       P13 Line 325,  delete “as expected, ”

-       P13 Line 327, a recently described novel species > a newly characterized (or described) species

-       P14 Line 340, for which strains > that

-       P15  Line 363, a relatively high level of similarity which was above 60% in both case (…) > a relatively high-level similarity of over 60% in both cases

-       P15 Line 378, “…after cultivation for 2 days on MEA Oxoid,…” > “…after cultivation for 2 d on MEA (Oxoid)…

Similarly, in scientific paper, xx days or xx minutes can be shortened to xx d, xx min.

Line 378-379, “…so we can assume it is linked to this specific condition.” > “…so it is assumed that spectrum discrimination is linked to this specific condition. “

-P15 Line 380-382, in the sentence of For instance, M. bainieri spectra at 2 days were on the left of the right quadrant while the spectra at 8 days were …

at 2 days > 2 d post incubation

at 8 days > 8 d post incubation

-        P15 Line 393-394, “…ranged between 90.48% to 100% with four out of five species yielding …” > …ranged from 90.48% to 100% where four out of five species were found to yield …

-        Line 396, as for “…ranging between 73.68% to 100%” : between > from

-        In the Discussion section, P16 Line 413-414,

 “…was 96.55% with 3.1% no identification and 0.35% wrongly identified spectra.> … was 96.55% with 3.1% of unidentified and 0.35% of mis-identified.

-        P16 Line 415-416, “… detailed in this study (Table 3) had between 80% to
100% of spectra correctly identified.
>  … examined under this study (Table 3) had correct identification rate ranging from 80% to 100%.

-        P16 Line 417-418,  “…among the 139 species, 109 of them yielded an overall correct identification rate of 100%, representing 78.42% of the species listed.> …109 out of 139 species yielded an overall correct identification rate of 100%, accounting for 78.42% of the species examined.

-        Line 418-419, … four species fell below … > the correct identification rates of four species fell below…

-        P17 Line 424-425,  “…can be difficult or even impossible to distinguish …> …it is difficult or even impossible to distinguish…

-        -     P17 Line 434-435, “…90.48% spectra were correctly assigned as expected which is promising.>

-             It is promising that 90.48% spectra were correctly assigned as expected.

-        -    P17 Line 435-437, “All spectra that were incorrectly identified corresponded to closely related species from the same genus and/or from the same species complex.” > The mis-identified spectra were ascribed to either the closely related species from the same genus or from the same species complex or both.

-        -P17 Line 437-439, in the sentence of “…spectra from C. snafimbriatum were identified as C. allicinum and C. macrocarpum which are closely related species within the same species complex”, it is suggested a comma should be put after C. macrocarpum and “which are” could be deleted. In addition, “two” could be added before “closely related species”.

-        -P17 Line 439,  “To address this, the database could be expanded in a future version to include more species from the C. herbarum complex, including…”

-        in a future version  > further; include  > encompass <to avoid the wording redundancy>

-        - P17 Line 471-472, “mycotoxin-producing species” > mycotoxin producers

2.   Formatting: for example, reference cited in the text—Leong et al, 2011 (P7 Line 170 of “Materials & Methods”).  To the best of our knowledge, the reference citation should follow certain format, either the sequential order or the alphabetic order.  

In addition, no link was created between the reference cited in the text of this manuscript and the references listed in the Section of References.

Author Response

Review report 1

Detail comments:

Minor issues:

  1. Vocabulary:

- P2 Line 58-59, in the sentence of “…rapid and accurate identification of fungi to the species level is crucial to ensure food safety…” : to—> at

Done (line 59)

- P2 Line 70-71, “…requiring special skills and knowledge to be applicable in routine in an industrial context” —> ”requiring special skills and knowledge to be applicable in routine examination at the industrial setting”.

Done (line 71)

- P2 Line 94, “…of technological interest and toxigenic ones” —> “…of technological and toxigenic interests”

Done (line 94)

- P3 Line 100-103, in the sentence of “…rapid and accurate identification of species within species complex using an updated spectral database could help distinguish species with diverse incidences and optimize prevention and control of fungal spoilage in food.”

help distinguish —> facilitate distinguishing;

optimize prevention —> boost the prevention

Done (line 101-102)

- P3 Line 104, the aim —> the goal

Done (line 103)

- P9 Line 266, ranging between 2 and 8 days —>ranging from 2 to 8 days

Done (line 265)

Line 267, “Positive and negative controls were made using respectively only reagents…” respectively only reagents —> the corresponding reagents only

Done (line 266)

Line 271, “…for those not in the database.” —> … for those not included in the database

Done (line 270)

- P13 Line 318, “were either not or incorrectly identified” —> were either unidentified or inappropriately identified.

Done (line 317)

- P13 Line 323, … not included in the database —> excluded in the database

We changed the term by “for those that were not part of the database” for better clarity (line 323).

- P13 Line 325, delete “as expected, ”

Done (line 325)

- P13 Line 327, a recently described novel species —> a newly characterized (or described) species

Done (line 327)

- P14 Line 340, for which strains —> that

Done (line 339)

- P15 Line 363, a relatively high level of similarity which was above 60% in both case (…) —> a relatively high-level similarity of over 60% in both cases

Done (line 362)

- P15 Line 378, “…after cultivation for 2 days on MEA Oxoid,…” —> “…after cultivation for 2 d on MEA (Oxoid)…”

Similarly, in scientific paper, xx days or xx minutes can be shortened to xx d, xx min.

Done (line 377)

Where appropriate, the term « days » has been changed to « d » throughout the manuscript.

Line 378-379, “…so we can assume it is linked to this specific condition.” —> “…so it is assumed that spectrum discrimination is linked to this specific condition. “

Done (line 377)

-P15 Line 380-382, in the sentence of “For instance, M. bainieri spectra at 2 days were on the left of the right quadrant while the spectra at 8 days were …”

at 2 days —> 2 d post incubation

at 8 days —> 8 d post incubation

Done (line 380-381)

- P15 Line 393-394, “…ranged between 90.48% to 100% with four out of five species yielding …” —> …ranged from 90.48% to 100% where four out of five species were found to yield …

Done (line 392-393)

- Line 396, as for “…ranging between 73.68% to 100%” : between —> from

Done (line 395)

- In the Discussion section, P16 Line 413-414,

“…was 96.55% with 3.1% no identification and 0.35% wrongly identified spectra.” —> … was 96.55% with 3.1% of unidentified and 0.35% of mis-identified.

Done (line 412)

- P16 Line 415-416, “… detailed in this study (Table 3) had between 80% to 100% of spectra correctly identified.” —> … examined under this study (Table 3) had correct identification rate ranging from 80% to 100%.

Done (line 413-414)

- P16 Line 417-418, “…among the 139 species, 109 of them yielded an overall correct identification rate of 100%, representing 78.42% of the species listed.” —> …109 out of 139 species yielded an overall correct identification rate of 100%, accounting for 78.42% of the species examined.

Done (line 414-415)

- Line 418-419, … four species fell below … —> the correct identification rates of four species fell below…

Done (line 416)

- P17 Line 424-425, “…can be difficult or even impossible to distinguish …” —> …it is difficult or even impossible to distinguish…

Done (line 422)

- - P17 Line 434-435, “…90.48% spectra were correctly assigned as expected which is promising.” ->

- It is promising that 90.48% spectra were correctly assigned as expected.

Done (line 432-433)

- - P17 Line 435-437, “All spectra that were incorrectly identified corresponded to closely related species from the same genus and/or from the same species complex.” —> The mis-identified spectra were ascribed to either the closely related species from the same genus or from the same species complex or both.

Done (line 433-434)

- -P17 Line 437-439, in the sentence of “…spectra from C. snafimbriatum were identified as C. allicinum and C. macrocarpum which are closely related species within the same species complex”, it is suggested a comma should be put after C. macrocarpum and “which are” could be deleted. In addition, “two” could be added before “closely related species”.

Done (line 436)

- -P17 Line 439, “To address this, the database could be expanded in a future version to include more species from the C. herbarum complex, including…”

- in a future version —> further; include —> encompass <to avoid the wording redundancy>

Done (line 438)

P17 Line 471-472, “mycotoxin-producing species” —> mycotoxin producers

Done (line 469)

  1. Formatting: for example, reference cited in the text—Leong et al, 2011 (P7 Line 170 of “Materials & Methods”). To the best of our knowledge, the reference citation should follow certain format, either the sequential order or the alphabetic order.

References in the text have been checked and updated when necessary (lines 169 and 653, 655)

In addition, no link was created between the reference cited in the text of this manuscript and the references listed in the Section of References.

To the best of our knowledge, links between cited references and references in the references section will be made after acceptance of the manuscript.

Reviewer 2 Report

The study focused on expanding the MALDI-TOF MS spectral database by acquiring additional spectra from spoilage fungi. Their contribution to the field is significant as it further complements the widely-used spectral database for rapid identification of spoilage fungi, which would help reduce food safety risk in the food industry. The paper gave a clear introduction to the current risks of food spoilage and how MALDI-TOF MS became a preferred method for identifying microorganisms, followed by setting the goal of expanding the spectral database. It clearly described what fungal species to add into the database and how to evaluate the identification accuracy by both an unsupervised method (t-SNE) and a supervised method (ASC algorithm with 5-fold CV). The authors presented the identification results and clusters with great concision by using tables and scatter plots, which makes it fairly easy to interpret and follow. They went into great depth when explaining the rationale for both correct identifications and misidentifications, followed by discussion on room for improvement and possible next steps in their research. Overall, this study has been conducted with high quality and value and the writing was impeccable.

The paper described how to expand the MALDI-TOF MS spectral database for identifying spoilage fungi and used both unsupervised and supervised algorithms to prove that MALDI-TOF MS spectra can be used to distinguish different fungal species. The paper was written with remarkable clarity and precision, and the high accuracy of identification was very encouraging.

Figure 1: It might be better to make the background white to make the dots more visible.

Line 511: There seems to be a typo here. Is it “the t-SNE”?

Author Response

Review report 2

Detail comments:

The paper described how to expand the MALDI-TOF MS spectral database for identifying spoilage fungi and used both unsupervised and supervised algorithms to prove that MALDI-TOF MS spectra can be used to distinguish different fungal species. The paper was written with remarkable clarity and precision, and the high accuracy of identification was very encouraging.

  • Figure 1: It might be better to make the background white to make the dots more visible.

The figure has been modified according to the reviewer’s comment.

  • Line 511: There seems to be a typo here. Is it “the t-SNE”?

Done (line 511)

Reviewer 3 Report

Scientific article contributes significantly to the development of advanced identification methods in microbiology. The authors performed substantial amount of research, and contributed to the identification and evaluation of the properties of food-relevant fungal species, as well as expanded the database of these types of microorganisms.

Science-intensive research using the newest approaches was carried out, and 380 yeast and mold strains belonging to 51 genera and 133 species were studied and included in the spectral database. The data obtained were analyzed, systematized and compared with the results of other researchers.

All the data obtained are clearly presented and also compared with the results of other studies. Overall, the article is highly relevant, scientifically sound, and advanced in this sector of microbiology.

Author Response

Thanks for your comments on the quality of the manuscript.